# Monitoring of Cow Location in a Barn by an Open-Source, Low-Cost, Low-Energy Bluetooth Tag System

**DOI:** 10.3390/s20143841

**Published:** 2020-07-09

**Authors:** Victor Bloch, Matti Pastell

**Affiliations:** Natural Resources Institute Luke (Finland), Latokartanonkaari 9, 00790 Helsinki, Finland; matti.pastell@luke.fi

**Keywords:** RSS localization, indoor positioning, Bluetooth low energy (BLE), cow location, barn environment

## Abstract

Indoor localization of dairy cows is important for cow behavior recognition and effective farm management. In this paper, we propose a low-cost system for low-accuracy cow localization based on the reception of signals sent by an acceleration measurement system using the Bluetooth Low Energy protocol. The system consists of low-cost tags and receiving stations. The tag specifications and the localization accuracy of the system were studied experimentally. The received signal strength propagation model and dependence on the tag orientation was studied in an open-space and a barn environment. Two experiments for the evaluation of localization accuracy were conducted with 35 and 19 cows for two days. The localization reference was achieved from feeding stations, a milking robot and videos of cows decoded manually. The localization accuracy (mean ± standard deviation) was 3.27 ± 2.11 m for the entire barn (10 × 40 m^2^) and 1.9 ± 0.67 m for a smaller area (4 × 5 m^2^). The system can be used for recognizing long-distance walking, crowded areas in the barn, e.g., queues to milking robots, and cow’s preferable locations. The estimated system cost was 500 + 20 × (cow number) € for one barn. The system has open-access software and detailed instructions for its installation and usage.

## 1. Introduction

Body location and its motions are important for farm animals’ well-being and efficiency monitoring [1]. Early alarms about dairy cow diseases, stresses and calving can be generated through analyzing untypical behavior and motions [2,3]. A variety of commercial systems based on tags with accelerometers provide the ability to monitor cow behavior derived from cow motions. However, these systems have a relatively high cost.

A dairy barn makes a challenging environment for localization due to the relatively high density of animals and the presence of metal structures. In order to monitor the indoor location of dairy cows, several technologies were used. Different ultra-wideband (UWB) systems were used in several studies. The Ubisense UWB system was used in [4] with 0.5–2 m accuracy depending on the position in the barn. The ability to detect the presence of cows in 0.5–3 m areas with a sensitivity of over 70% using UWB-based Gea CowView with 3–4 years battery lifetime was reported in [5]; the system reached the accuracy of 16 cm with static tags with worst case positioning error of several meters when tags were attached to cows. A custom setup based on Decawave UWB DW1000 chip reached a static accuracy of 38 cm [2]. Commercial wireless local area network (WLAN) systems with 2 m of accuracy and 30 day battery lifetime was used in [3], and with 3 m of accuracy and 1-day battery lifetime was used in [6]. Image processing was used for cattle motion and behavior monitoring [7]. A self-designed Bluetooth-based system was used by [8] with 4.2 m of accuracy. Outdoor animal location was monitored by a system combining GPS tags for global positioning and BLE tags for the identification and approximate localization of animals equipped with BLE tags [9], and a BLE-based system [10].

The advantage of the Bluetooth Low Energy (BLE) technology is in long battery life and low sensor price. Nevertheless, the received signal strength (RSS) of BLE tags used for localization is noisy [11]. In addition, a number of factors influencing the RSS and the accuracy of the tag localization calculated by trilateration [12] must be taken into account: irregularity of the RSS propagation model [11], direction of the tag headed to the receiving station [13], cow body decreasing the RSS, barn obstacles reflecting and increasing the RSS [14] etc., which, in general, according to the estimation of [15], limits the accuracy of this method to 2.7 m. To compensate for these factors, various methods were applied. The RSS propagation model was studied in [11,16,17]. To filter the tag positions inside a barn, median and extended Kalman filters [4,18], the Viterbi algorithm [19,20] and Bayesian filters [21] were used. The BLE signal channels were treated separately in [22,23]. The developed signal propagation model included the geometry and material properties of the structures of the tag environment as well as the mapping of RSS depending on what the tag orientation was used in [16,24]. The RSS decreasing while passing through the dairy cow’s body was modeled [25], and human-body signal shadowing was mitigated [26]. Algorithms for learning the environment features were used for the localization [27]. Information about the environment structure was used to eliminate impossible locations [8,19,28]. Instead of describing the environment features, a preliminary mapping (fingerprints) of RSS from several points in the environment was performed [10,18,29], though this method required time-consuming mapping, and its accuracy can be restricted by the resolution of the sampled points.

The goal of this study was to design a system for cow localization in barn environment and estimate its accuracy. We developed a system for cow location monitoring. The system includes low-cost tags attached to cow collars, which sends signals by the BLE protocol to low-cost signal receiving stations installed in a barn similar to [30]. In this system, the RSS was received as an accompanying effect of the cow motion accelerations monitoring measured by the tags. The RSS was filtered by a moving average filter. The influence of the tag orientation on the localization was studied. The tag location was filtered by the Viterbi algorithm. The system accuracy was tested in a research barn. The data and algorithms used in this study are attached in Appendix A.

The paper is organized as follows. Section 2 describes the structure of the localization system, experimental environment, methods used for the localization and conducted localization experiments. Section 3 describes the tag RSS features studied in the experiments. Section 4 presents the results of the experiments. Section 5 specifies the localization system accuracy, analyzes the experimental results and compares them with other studies. Section 6 summarizes the findings and proposes topics for the future research.

## 2. Localization System and Accuracy

### 2.1. System Design

The system was developed as part of a project aiming to develop a low-cost open-source system for monitoring the behavior of dairy cows using accelerometers and positioning. This paper focuses on using tags for cow localization. An open-source BLE tag (RuuviTag, Ruuvi Innovations, Porvoo, Finland) based on the nRF52832 chip (Nordic Semiconductor, Trondheim, Norway) was chosen as the platform. New firmware for measuring acceleration at 25 Hz and advertising the data was developed.

The tags packed in plastic boxes and adjusted to the cow collars on a side by a Velcro belt (Figure 1) measured the acceleration and sent the measurements as advertising data (24-bytes packets) using the BLE 4.2 protocol. The frequency of message sending was 5 Hz. The messages from the tags were received by 10 receiving stations, which were single-board computers with Bluetooth antennae (Raspberry Pi 3 B+, Raspberry Pi Foundation, UK). The stations were packed in hermetic cases with heat dissipation ribs and installed on barn constructions on the height of 3–5 m (Figure 1c). They were evenly distributed in the barn (Figure 2a) to minimize the maximal distance to tags. The stations recorded the RSS, tag accelerations and receiving time. The data was stored on the station memory and was sent via a local network maintained by a router (EA7500, Linksys) by a message queuing protocol (ZeroMQ, iMatix Corporation). A PC (Intel^®^ Core™ i7-9750H, CPU 2.6GHz, RAM 16GB) received the messages from the microcontrollers and stored the raw data in CSV files. The data managing programs were written on C++ for the stations and C# (Microsoft, Redmond, WA, USA) for the PC.

### 2.2. Barn and Cows

A research barn (University of Helsinki, Viikki campus) was used for the development and validation of the system. The barn consisted of a 9.8 × 42 m^2^ area (Figure 2a) with 46 lying pens and included 24 feed intake measuring stations (Hokofarm, Marknesse, the Netherlands) and a milking robot (Astronaut, Lely, Maassluis, the Netherlands). The barn was equipped with five constantly recording cameras covering major part of the area. A group of 35 Ayrshire cows were used in the experiments. The cows were housed in a free stall barn (Figure 2b) during the lactation period.

### 2.3. Localization

Considering the large number of factors influencing the RSS in the barn environment (diversity of tag and receiving stations features, tag orientation, tag occlusions depending on cow movements, multiple metal structures in the barn), we assumed that RSS fingerprint mapping or using path loss models based on the barn structure description are impractical for this application. Hence, only the RSS propagation model created in a tag feature experiment (described in Section 3.1) was used.

The RSS propagation model (path loss log-distance model [31]) describing the dependence between the RSS and the distance to a tag was formulated by Equation (1),
(1)$$RSS=-10\cdot n\cdot \log\left(D\right)+A_{0},$$
where *D* was the distance between the receiving station and the tag, and *n* and *A_0_* are the path loss exponent and the path loss constant fitted by linear regression.

Simple geometric trilateration for the tag location calculation was impractical because of high RSS noise. Hence, 10 receiving stations achieved redundant information about the tag location. The tag localization was equivalent to the solution of an optimization problem finding the tag position with minimal RSS errors for all receiving stations. The optimization problem was defined as the minimization of the localization error function *Err*(*x,y*):(2)$$\underset{x,y}{\min}Err\left(x,y\right)=\underset{\begin{array}{c} x,y\\ s.t.\\ 0\leq x\leq 9.8\\ 3\leq y\leq 42 \end{array}}{\min}\overset{10}{\underset{i=1}{\sum }}{\left(RSS_{mes,i}-RSS_{pm}{\left(x,y\right)}_{i}\right)}^{2},$$
where (*x,y*) were the tag location in the barn coordinates (Figure 2a), *RSS_mes,i_* was RSS measured by the receiving station *i*, *RSS_pm_(x,y)_i_* was RSS calculated according to the propagation model (Equation (1)) depending on the distance between the receiving station *i* and the tag located in the point (*x,y*). This optimization problem had two variables, which were the tag coordinates in the barn. It was assumed that the height (Z coordinate) of the tag was 1.5 m and the dependence between the tag RSS and tag orientation was not taken into account.

However, during the tag characteristics experiments, we found significant variability between the RSS of different tags and receiving stations. Hence, the difference between the measured and the calculated values of RSS was expected to reflect the actual distance between the tag and the receiving station incorrectly. Assuming that the RSS propagation model for all tags has same curve shape (coefficient *n* in Equation (1)) but different shift (coefficient *A_0_* in Equation (1)), we canceled this difference by the shifting factors. Then, the localization error function was revised as follows:(3)$$\underset{x,y}{\min}Err\left(x,y\right)=\underset{\begin{array}{c} x,y\\ s.t.\\ 0\leq x\leq 9.8\\ 3\leq y\leq 42 \end{array}}{\min}\overset{10}{\underset{i=1}{\sum }}{\left(RSS_{mes,i}-N_{mes}-\left(RSS_{pm}{\left(x,y\right)}_{i}-N_{pm}\right)\right)}^{2},\;N_{mes}=mean\left(RSS_{mes,i}\right),\;N_{pm}=mean\left(RSS{\left(x,y\right)}_{i}\right),$$
where *N_mes_* was a shifting factor for the measured RSS, and *N_pm_* was a shifting factor for the RSS calculated according to the propagation model.

To make the localization process more practical, the solution for the optimization problem was replaced by finding the best fitting between the measured RSS and the RSS calculated in predefined mapping points and stored in a lookup table. The reasons were the following:The expected accuracy of the method was lower than the barn mapping points resolution; hence, a continuous solution for the optimization problem was senseless.A finite number of locations allows to make further analysis of the found locations and filtering them by the Viterbi algorithm, taking into account the barn structure features.Comparison of the calculated location with the reference location achieved from videos (required intensive manual work) was more convenient with predefined points.Optimization problem solution was in general a more time-consuming problem than a search in a lookup table. In addition, the nonlinearity of the cost function could cause multiple minima, requiring additional analysis of the function.

The number of the predefined location points (*x_k_,y_k_*) was 235 for entire barn and 95 for the waiting yard. They were distributed over the barn area with a resolution of about 1 m (Figure 2a) for entire barn and 0.5 m for the waiting yard. To create the RSS mapping of the barn, the RSS values for each station were calculated in each location point using the RSS propagation model derived from experiments and stored in a lookup table *RSS_map_*.

### 2.4. Tag Location Filtering

The level of noise of RSS and, as a result, of the tag location, makes filtering the tag location by average or medial filters inefficient. The Viterbi algorithm using the barn structure and including a Markov chain describing the probability of passages between the location points was used. 

The observed states were the RSS values. The hidden states were the location in the barn mapping points. The emission matrix (P) in the Viterbi algorithm (fitting between the hidden and observed states) was replaced by a rule for calculating the probability to be located in all the mapping points. The probability of being in a point *k*, *P_k_*, depended on the error between the *RSS_mes_* measured by the receiving stations and *RSS(x_k_,y_k_)* precalculated for all mapping points as defined in Equation (4):(4)$$P_{k}=\frac{1}{Err\left(x_{k},y_{k}\right)},\;k=1..235$$

To calculate the transition matrix (TM), taking into account the distance between the location points and obstacles, a map of relevant barn structures was created based on a provided barn drawing and manual measurements. TM was built according to the following rules:

*TM_i,i_* = 1 – probability to stay in the same location;

*TM_i,j_* = 0 – if there is an obstacle between the locations *i* and *j,* or distance between them *dist(i,j)* is more than 3 m;

*TM_i,j_* = 1/(1 + *dist(i,j)*) – if the previous conditions are false.

The transition matrix was normalized to provide the sum of the probabilities equal to 1.

The original RSS data were obtained with a sampling time of 0.2 s. This caused impossible passages between the location points during a single sampling time. To overcome this effect and reduce the computational effort, the RSS data was decreased by folding intervals of 5 s into a single sampling point, where the RSS value was taken as the average value on the intervals.

The entire process for the tag localization is presented in Figure 3. 

### 2.5. Application of the Localization System

To estimate possible areas of the system applicability, the location of the cows from the entire barn experiment was monitored during one week with the help of the localization system. A map with the distribution of time spent in each location point during this period was created for all the cows.

### 2.6. Estimation of System Accuracy

To estimate the accuracy of the cow localization system, two experiments with different receiving station arrangements were conducted in order to find the influence of the average distance between the receiving stations (resulting in the number of receiving station covering the barn area) on the method accuracy. In experiment 1, the entire barn experiment included 10 receiving stations distributed over the entire area with the average distance of 10 m between the neighboring stations, as shown in Figure 2a. This experiment was conducted with 35 cows wearing 12 tags in groups during four periods of about two days. Experiment 2 included 7 receiving stations distributed in the waiting area in front of the milking robot with the average distance between the neighboring stations of 2 m, as shown in Figure 2a. This experiment was conducted with 19 cows wearing the tags for two days. 

The reference cow location was measured by the following systems in the barn. Location during eating and drinking was recorded by the feeding stations. Location during milking was recorded by the milking robot. Locations during lying in cubicles, staying and walking in the alleys were recorded by the barn cameras (Figure 2b) and classified manually. The location of a cow in a specific barn’s point was registered if this point was the closest to the cow collar for more than five seconds. A continuous walk between two points distanced by more than 5 m was also registered. The clocks of all the systems were synchronized with the receiving stations with accuracy better than 1 s. 

## 3. Tag RSS Features

### 3.1. Tag Characteristics Experiment Setup

The RSS characteristics were measured in two experiments: RSS propagation and RSS-tag orientation. To maximally eliminate the influence of signal reflecting objects on the RSS, the experiments were conducted in the open space (parking lot) on the height of 1.8 m above the ground (Figure 4). The closest metal and concrete objects were distanced by at least 20 m (excluding asphalt ground and lamp posts).

The RuuviTag without enclosure with its coordinate system is presented in Figure 4a. In the RSS propagation experiments the tag was oriented in a way that the receiving station was on the X axis, and the tag Z axis was upwards. For each distance, the recording of RSS was conducted during 120 s with frequency of 5 Hz (totally 600 samplings). To achieve an RSS value representing the distance between the tag and the receiving station, the RSS samplings were averaged. In all the experiments, the tags were put in an enclosure later used in the barn.

The RSS propagation model (coefficients *n* and *A_0_*) based on data achieved from the open-space experiment was set as the propagation model of the localization system and used in barn experiments. The open-space experiment was conducted to achieve the absolute features of the RSS tags and the receiving stations. In addition, we conducted barn experiments, which were intended to demonstrate a possible influence of the environment in specific conditions.

### 3.2. RSS Propagation Model and RSS Diversity

To find the propagation model of the tag RSS and the receiving station, an experiment measuring the RSS on distances from 1 to 15 m with the step of 1 m was conducted. To investigate possible factors influencing the variability of the tag RSS, the propagation model experiment was conducted with 10 tags and 10 receiving stations. Additional illustration of the propagation model was made from data collected in barn conditions.

### 3.3. RSS-Tag Orientation Dependence

The diversity in the RSS values caused by the tag orientation represents a significant factor for the uncertainty in the tag localization; hence, it was studied carefully. To find the dependence between the tag RSS and the tag orientation relative to a receiving station, an experiment with different tag orientations was conducted. The tag was rotated around ground Z and Y axes by angles *θ* and *φ* measured by the gimbal shown in Figure 4b with resolution 45° in ranges [0:135°] and [0:315°] respectively (totally 32 positions) and the initial position as in the experiments for RSS propagation. It was assumed that the RSS does not depend on the rotation around the line connecting the tag and the receiving station; hence, rotation around the third axis was not considered. The experiment was conducted with three tags and 10 receiving stations in the open space. To investigate the influence of the barn environment on the RSS-tag orientation dependence, an additional similar experiment with six tags and three receiving stations was conducted in the barn. 

### 3.4. RSS Filtering 

Three filters for filtering the RSS were tested: mean, median and Kalman. The dynamics of cow motions was considered to adjust the filter parameters. Since cows in a barn are located in same places most of the time (walking between different locations takes less than 5% of time according to our observations) and the expected accuracy of the method does not allow for tracking a cow’s walking, the filters have to eliminate the motions with a characteristic time of less than several seconds. Therefore, the filtering windows of 2, 10, 60 and 300 s were taken for the mean and median filters, and the covariance of the process noise (parameter q of the matrix Q in Equation (5)) was taken with values 10^−3^, 10^−5^, 10^−7^ and 10^−9^ for the Kalman filter. To compare the performance of the filters, the total average localization accuracy using all available reference data was calculated for each filter. The structure of the Kalman filter was as follows:(5)$$F=\left(\begin{array}{cc} 1 & \Delta t\\ 0 & 1 \end{array}\right),\;B=\left(\begin{array}{c} 0\\ 0 \end{array}\right),\;u=0,\;Q=\left(\begin{array}{cc} q & 0\\ 0 & q \end{array}\right),\;H=\left(\begin{array}{c} 1\\ 0 \end{array}\right),R=1,$$
where Δ*t* = 0.2 s was the signal sampling time, and *q* was fitted in testing.

### 3.5. Battery Life-Time Estimation

The power consumption of tags was evaluated by a power profiler kit (nRF52, Nordic Semiconductor). During the profiling, a tag was connected to the kit’s power source. The sampling cycle performed at the frequency of 5 Hz consisted of the following operations: acceleration sampling performed at 25 Hz, receiving two advertisements and sending the data package.

## 4. Results

### 4.1. Tag RSS Features

Typical RSS and its distribution for the open-space (for four distances between the tag and the receiving station) and barn environments (while the cow with the attached tag was sleeping) are presented in Figure 5.

An example of the RSS sent from one tag and measured by one receiving station is presented in Figure 6a. Each RSS sampling is represented by a dot, the average of RSS for specific distances is represented by a straight line, and the RSS propagation model described by the Equation (1) with coefficients calculated by regression is represented by a dashed line.

The data collected in the RSS propagation model experiment from all the combinations of 10 tags and the 10 receiving stations with the average line (solid line) is presented in Figure 6b. The total RSS propagation model described by the Equation (1) achieved by regression with R^2^ = 0.67 (dashed line) had *A_0_* = −48.77 ± 4.070 dB, *n* = 0.84 ± 0.119 (mean ± standard deviation (STD)). This model (RSS = −10 ∙ 0.84 ∙ log(D) − 48.77) was later used in the barn experiments. 

The lines representing the average of all RSS values achieved from all the 10 tags Figure 6c and by all the 10 receiving station Figure 6d are shown with the same total averaging line (solid). To estimate the variability of the RSS of each tag received by all the stations shown in Figure 6c and the variability of the RSS of stations receiving the signal from all the tags shown in Figure 6d, the average values of the standard deviations for each tag and each receiving station were calculated. The STD for tags was 2.7 dB, while for the receiving stations the STD was 4.14 dB.

An illustration of the RSS propagation measured in the barn is shown in Figure 7. Two sets of RSS during cows walking with constant speed on a straight line measured by a receiving station located on this line are presented by dots and asterisks.

The data achieved in the RSS orientation experiment from all the combinations of the tags and the receiving stations with the average lines is presented in Figure 8 for the open space (a) and barn (b). The mean RSS deviation in the open space was about 17 dB, while in the barn it was about 6 dB.

Current consumption during a typical cycle of tag activity is shown in Figure 9. The average current consumption was 12.9 µA during an interval of 60 s with a maximal current value of 11.4 mA. Assuming a nominal battery voltage of 3 V, the battery capacity of 1 Ah is sufficient for 1068 days or 2.9 years.

### 4.2. Tag Localization

The results of applying different filters on the RSS are presented in Table 1, with examples of filtered signals presented in Figure 10.

An example of the tag localization by the Viterbi algorithm compared to the actual locations is shown in Figure 11a for one tag during one day in experiment 1. The graph represents the barn mapping points (Y-axis) in which the tag was located and calculated during the day.

The performance of the Viterbi algorithm is demonstrated in four specific moments in Figure 11b. The heat maps represent the probability of the tag localization calculated in each mapping point and based on the measured RSS and the probability to move to other location points defined by the barn structure (Equation (4)). Since the Viterbi algorithm for each sampling moment takes into account locations at the nearby moments (minimizing the total location error for the entire time interval), the calculated locations (circle) are not necessarily coincident with the points with the maximal location probability, which is not necessarily close to the actual tag location (asterisk) because of the high uncertainty of the RSS.

The estimation of the localization accuracy is presented in Figure 12 for experiment 1 (entire barn) (a) and experiment 2 (b) (waiting yard). The localization error for each cow is presented by boxplots. The total average location error (mean ± STD) was 3.27 ± 2.11 m in experiment 1 and 1.9 ± 0.67 m in experiment 2. The numbers of the tags attached to different cows are denoted above the boxplots. The cumulative percentage of the location error for each cow is presented in Figure 12c,d, with the average cumulative percentage represented by the solid line.

An example of the application of the localization system for monitoring the cows’ locations for one cow and the mapping of total preferable locations in the barn are presented in Figure 13. The relative frequencies of locating in the mapping points calculated by the localization system in experiment 1 during 7 days are presented in Figure 13a for one cow (number 2 from the list in Figure 12a) and in Figure 13b for all observed cows in the barn. The time distribution of the cow’s location in the barn parts is presented in a histogram in Figure 13c.

## 5. Discussion

The location accuracy of the system covering the entire barn was 3.27 m and 1.9 m for the smaller area (4 × 5 m^2^), which makes the system much less accurate than systems based on UWB technology reaching 0.4–2 m of accuracy in entire barns [2,4,5], but with similar accuracy as the research system based on BLE, such as 4.2 m at [8] in the cow barn environment and 2.4 m (for 90% of tested cases) at [22], 3.8 m at [26] and 2.3 m at [16] in the office environment. The developed localization system can be used to recognize preferable cow locations, such as areas for lying (Figure 13a), feeding and crowded locations, such as queuing to the milking robot (Figure 13b) and a rough distribution of the cow activity (Figure 13c). However, it does not grant the ability to track the cows’ walking and time spent in more specific locations, such as individual stalls or feeders.

The Viterbi algorithm was chosen among other filtering methods (fingerprint, particle filter, Kalman filter) because of the following reasons: it has a low computation time; the location computation is based on the entire time interval, not only on the previous sampling; it uses additional information about cow behavior and the barn structure (which can be accessible for commercial barns); it does not require reference collecting for learning.

The RSS of the RuuviTag studied in this research had a relatively high noise level and non-gaussian distribution (Figure 5a) similar to [12,27] and others. Particularly, it was caused by the aggregation of RSS received by different channels (37, 38 and 39, separated by [26]), which can be seen in the first image in Figure 5a. However, in the barn environment, the RSS distribution was closer to gaussian (Figure 5b). One possible reason could be multiple reflections of the signal from randomly distributed obstacles, which can produce resultant signals with higher randomization.

The variability of the RSS for the considered tags and receiving stations had several origins. The deviation from the analytical line for the RSS propagation model (Figure 6a) can be explained by the high sensitivity of the RSS to the objects reflecting radio signals [14] (such as ground and distant objects which are hard to eliminate during this RSS propagation model experiment) and tag directional RSS variability combined with inaccuracies in the tag direction during the experiment. Trials to create the barn RSS map were conducted during the research. However, this action was extremely time consuming and inconvenient, even for rougher point resolution. In addition, considering the high RSS variability, uncertain factors such as blocking the signal by other cows, or cows bending necks, it was concluded that creating a barn RSS map was impractical, especially for commercial application.

Possible reason for the RSS variability between different tags and receiving stations (Figure 6b) can be the variability in features of the antennae of the tags and the receiving stations. When separately drawing the average RSS propagation lines for each tag received by all the stations (Figure 6c), relatively small variability between the tags can be seen, though the average STD for each tag was high, meaning that the signal emitted by a tag was received by different stations with significantly different RSS. Average RSS propagation lines for each station receiving signals from all the tags (Figure 6d) show that different stations receive signals from the same tags with high variability. The average STD for each station was lower than the average STD for the tags. This can show that each receiving station measures the RSS from all the tags with lower variability, while, on average, different stations measure different RSS sent by the same tags. In general, this can be used for calibrating the RSS propagation model for each station, which could decrease the RSS uncertainty. Nevertheless, total high RSS variability in the open-space experiment makes it impractical to perform additional experiments when their results are used in the barn environment with significantly higher uncertainties.

To simplify the usage of the system in commercial barns, the average RSS propagation model achieved in the experiment (Figure 6b) was used. In general, the actual measured RSS does not fit the model, which can be observed in the RSS propagation model experiment (Figure 6c,d) and the RSS measured in the barn (Figure 7). Nevertheless, the relative RSS (fixed by a shifting factor) can be close to the model line, which justifies the usage of the shifted localization error function (Equation (3)). The average localization accuracy with the unshifted localization error function (Equation (2)) was 4.1 m.

The variability of the RSS depending on tag orientation is relatively high (Figure 8a); on average, it is 17 dB, while, for a specific tag, it can reach 22 dB, which is similar to [13]. In general, assuming that the RSS directionality is known and unchanging, and the tag orientation can be defined by accelerometers and a compass (or when a compass is missing, the yaw angle can be treated as a variable in the optimal location fitting), the RSS uncertainty caused by the tag direction can be eliminated. This strategy was applied in the current study, though the average localization accuracy (4.4 m) was worse than when the RSS directionality was not used. A possible explanation can be the same as the explanation for the effect observed in Figure 5b: signal reflections from multiple objects distributed isotropically in the barn environment act as a filter decreasing the RSS variability caused by the tag direction. This assumption corresponds with the RSS-tag orientation experiment presented in Figure 8b, where the average deviation of the RSS for different orientations in the barn experiment is about 6 dB.

According to testing different RSS filters and their parameters, they do not have a significant influence on the accuracy of the localization method (in the considered range of parameters). This can be explained by further location filtering by the Viterbi algorithm, which compensates the uncertainties staying after the RSS filtering. The average filter with a 10 s window, providing better performance and the simplest implementation, was used. In this study, the simplest version of the Kalman filter unfitted to the RSS features was tested.

The accuracy of the localization method depends on the way for the receiving stations positioning. According to the two experiments conducted in the study, the accuracy is about half of the minimal distance between the receiving stations. Hence, the number and location of the stations must be fitted to the area of interest and the type of application.

The main reason for the low accuracy of the developed system based on RSS received by BLE technology was the high level of noise caused by the tag sensitivity on its position and complex barn environment. Further improvement of the localization algorithm can be done in the following parts of the system. The RSS noise can be decreased by the separation of the BLE signal channels, as was done in [26] and [22]. The Viterbi algorithm can be improved by more detailed description of the cow behavior by the Markov chain, changing a method for calculation of the location probability in the mapping points and using additional information about the cow behavior measured by other sensors (such as cow body position estimated with the help of tag accelerometers) similar to [16]. To filter the RSS, a more advanced Kalman filter fitted to the RSS features or another filtering method can be found.

## 6. Conclusions

The low-cost indoor localization system based on BLE tags and receiving stations provides an ability to monitor an approximate cow location with an accuracy of 3.27 ± 2.11 m in a barn (10 × 40 m^2^). It can be used for the following barn applications: detecting abnormally long walks during heating, detecting the queue to the milking robot, measuring the time spent indoor and on pasture. The applicability of the system for recognizing unusual behavior will be tested in large-scale experiments. The advantages of the proposed localization system relative to existing commercial systems is in its low cost and long battery life.

Unlike the majority of existing BLE-based system research, the proposed system was designed for animal tracking in a barn environment characterized by multiple objects reflecting and absorbing the BLE signal. Considering the similar systems [8], our system has better accuracy, and the RSS features of the tags in the barn environment were better studied. In future research, the accuracy of the localization can be improved by using the methods developed for an indoor environment and further research of the RSS features.

## Figures and Tables

**Figure 1 sensors-20-03841-f001:**
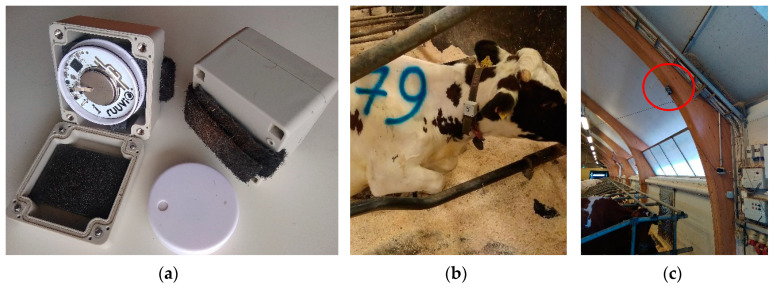
Components of the location and acceleration measuring system installed in a barn: RuuviTag inside a protecting plastic box (**a**), tag on the cow collar (**b**), receiving station installed on a barn construction (**c**).

**Figure 2 sensors-20-03841-f002:**
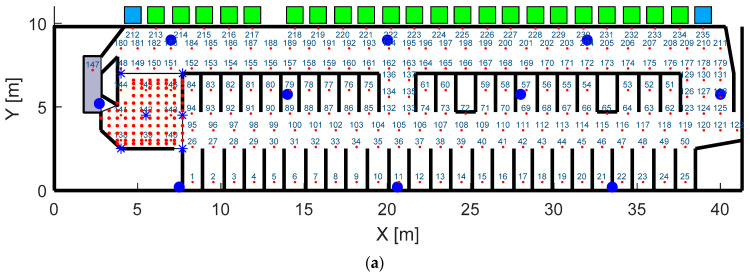
Map of the entire barn (**a**) with the waiting yard in front of the milking robot shaded by yellow. Squares represent feeders and water troughs. Grey rectangle represents the milking robot. Large blue dots and blue asterisks represent receiving stations in barn experiments. Small red dots and red asterisks represent the mapping points used in barn experiments. The general view of the barn from one of the video cameras is given in (**b**).

**Figure 3 sensors-20-03841-f003:**
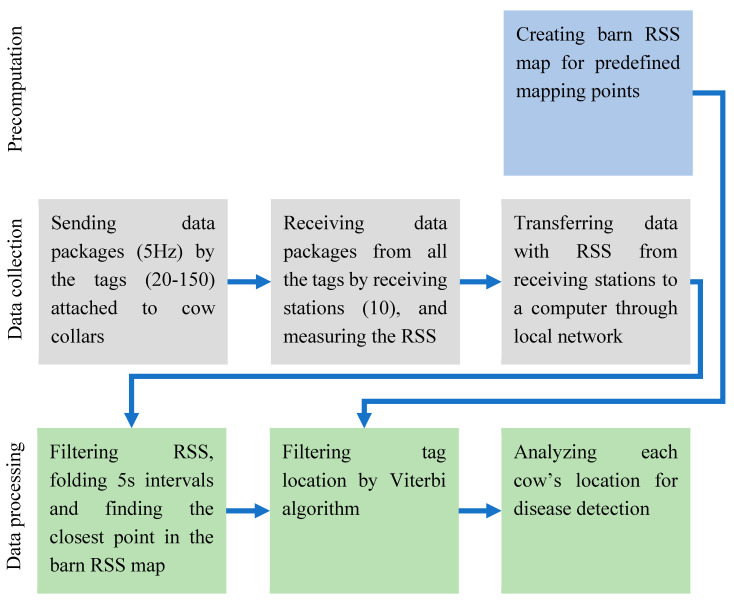
Cow localization flow chart.

**Figure 4 sensors-20-03841-f004:**
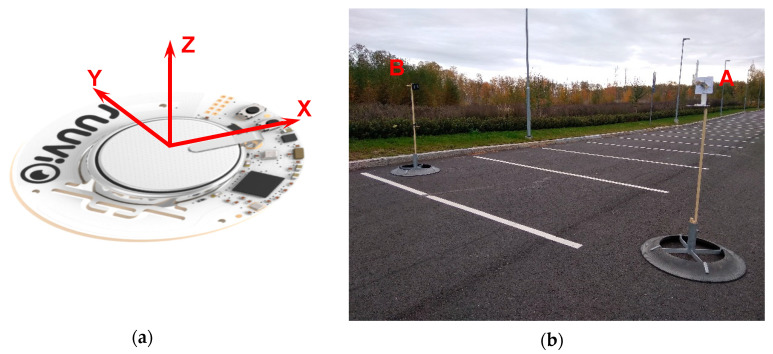
RuuviTag coordinate system (**a**). Setup of the propagation model and orientation experiment (**b**) with RuuviTag on a gimbal (A) and receiving station (B).

**Figure 5 sensors-20-03841-f005:**
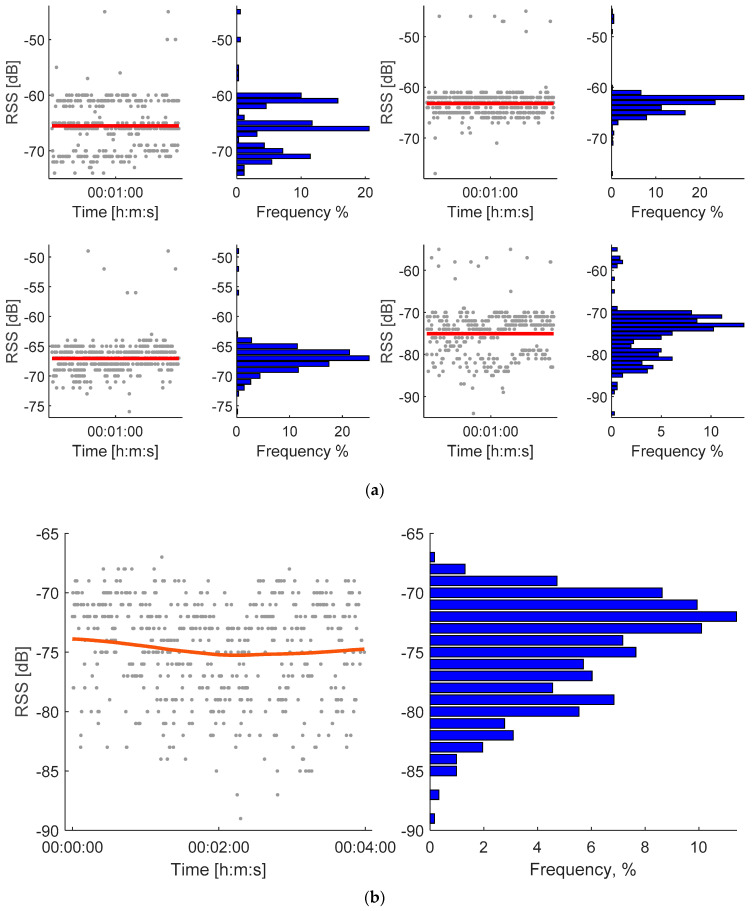
Typical received signal strength (RSS) during the measurement time and its frequency distribution in histogram for the open-space environment (**a**) and the barn environment (**b**). The distances between the tag and the receiving stations were 5, 10, 15 and 20 m accordingly in the open space (**a**) and 10 m in the barn (**b**). The signal was averaged similar to the RSS propagation experiment (**a**) and filtered similar to the actual RSS measuring in barn (**b**).

**Figure 6 sensors-20-03841-f006:**
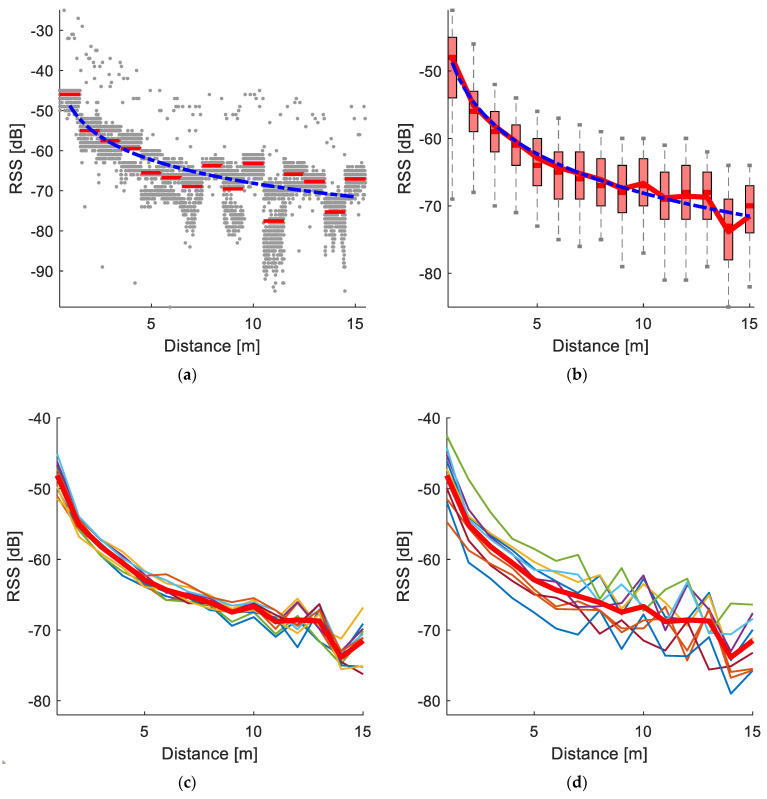
Results of the RSS propagation model experiment. A single experiment with one tag and one receiving station (**a**) with RSS samples is represented by the dots, average for each distance (red lines) and propagation model are described by Equation (1) (dashed line). All the combinations of the 10 tags and the 10 receiving stations with the average line (solid), propagation model (dashed line) and RSS deviation (boxplots) are presented in (**b**). The lines represent the average of all RSS values achieved from all the 10 tags (**c**) and by all the 10 receiving stations (**d**) with the same total average line (solid).

**Figure 7 sensors-20-03841-f007:**
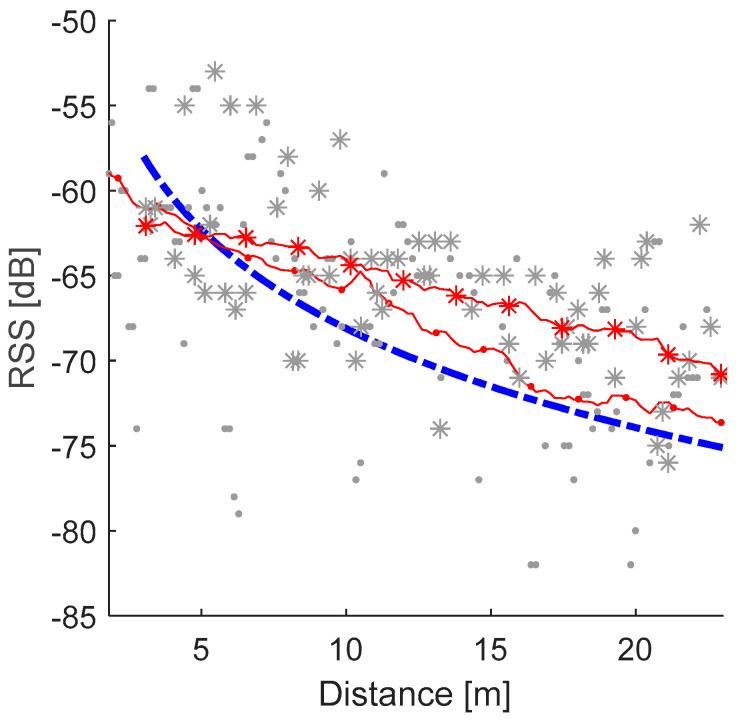
Relation between the RSS propagation model (dashed line) and RSS measured for two cows are represented by dots and asterisks (lines representing a filtered signal) walking with constant speed on a straight line in barn environment.

**Figure 8 sensors-20-03841-f008:**
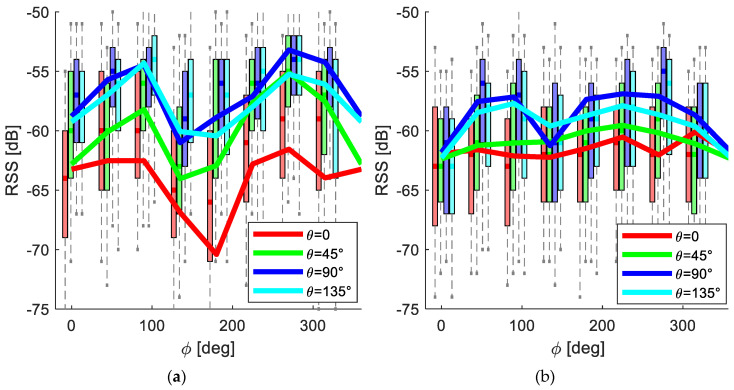
Results of the RSS-orientation experiment in the open space (**a**) and in the barn (**b**). The solid lines represent the average for each value of *θ*, and the boxplots represent the RSS deviation of the line with the corresponding color.

**Figure 9 sensors-20-03841-f009:**
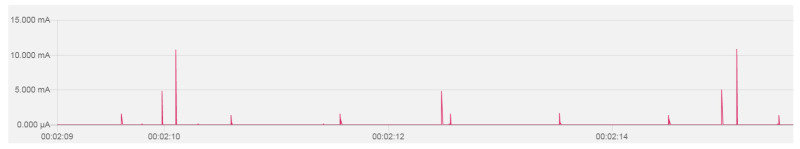
A typical cycle (5Hz) of tag current consumption, including five data samplings and data package sending.

**Figure 10 sensors-20-03841-f010:**
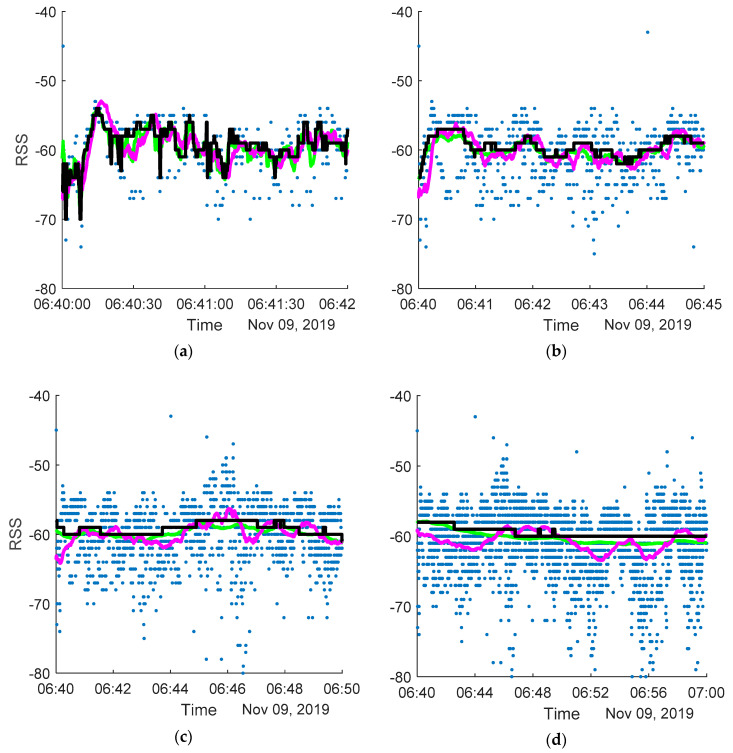
Filtering the RSS using different filters (green—mean filter, black—median filter, purple—Kalman filter) with different parameters: WS = 2 s, q = 10^−3^ in (**a**), WS = 10 s, q = 10^−5^ in (**b**), WS = 60 s, q = 10^−7^ in (**c**) and WS = 300 s, q = 10^−9^ in (**d**).

**Figure 11 sensors-20-03841-f011:**
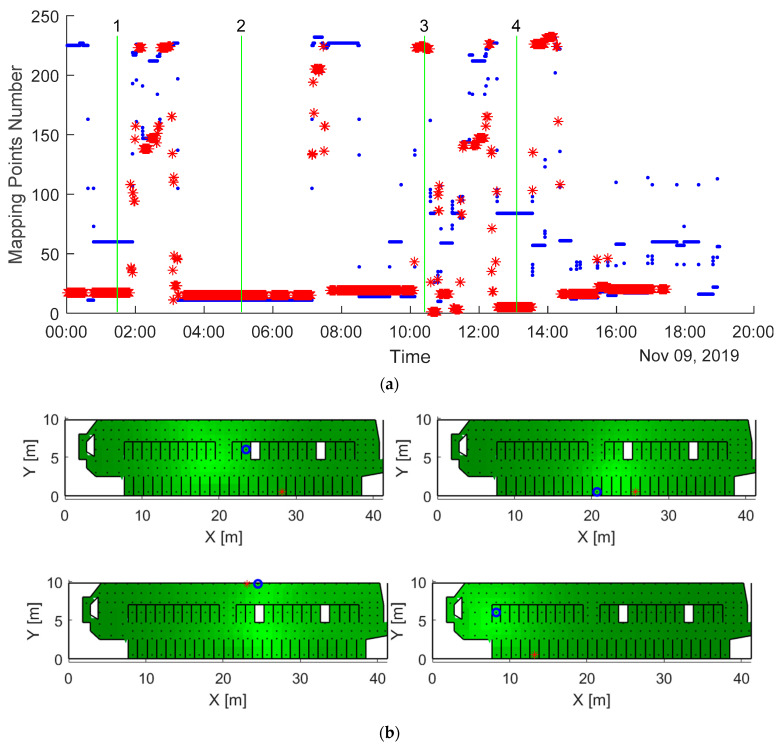
Comparison between the calculated (blue dots) and reference (red asterisks) tag locations during a day (**a**). Sequence of the barn maps with tag localization (**b**) corresponds to the moments signed on (**a**) by horizontal lines 1, 2, 3 and 4 respectively. The green gradient in the heat maps in (**b**) represents the probability of the tag localization in the mapping points in four instances indicated as green lines in the panel (the lighter the color, the higher the probability). Blue circles represent the calculated tag location, while red asterisks represent the reference.

**Figure 12 sensors-20-03841-f012:**
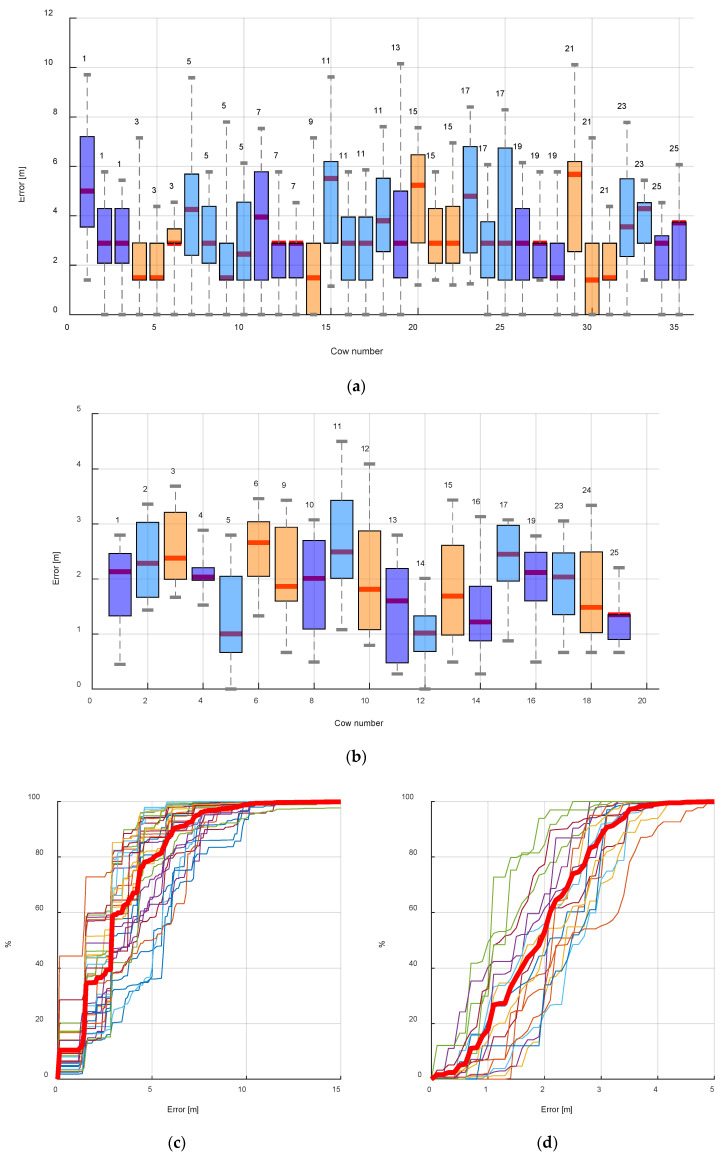
Total localization accuracy represented by the boxplot for each cow and tags (signed above the boxplot) and average cumulative accuracy for the entire barn (**a**) and (**c**) and waiting yard (**b**) and the experiments (**d**).

**Figure 13 sensors-20-03841-f013:**
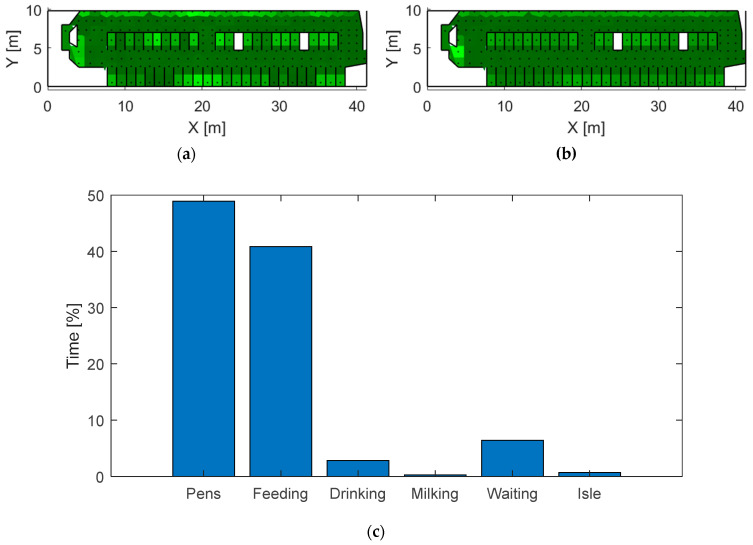
Relative frequencies of locating in the barn mapping points for one cow (**a**) and all cows (**b**) calculated by the localization system. The histogram shows the time distribution of the cow’s preferable locations (**c**).

**Table 1 sensors-20-03841-t001:** Average tag localization accuracy calculated for RSS filtered by different filters (mean ± STD) with a different window size (WS) and coefficient q for the Kalman filter.

Accuracy [m]	WS = 2 s, q = 10^−3^	WS = 10 s, q = 10^−5^	WS = 60 s, q = 10^−7^	WS = 300 s, q = 10^−9^
Mean filter	3.33 ± 2.19	3.27 ± 2.21	3.31 ± 2.12	3.67 ± 2.55
Median filter	4.13 ± 2.83	3.58 ± 2.45	3.60 ± 2.47	4.12 ± 2.89
Kalman filter	3.34 ± 4.32	3.35 ± 2.19	3.34 ± 2.18	3.42 ± 2.14

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
