# Peer review of "Monitoring of Cow Location in a Barn by an Open-Source, Low-Cost, Low-Energy Bluetooth Tag System"

_sensors, 2020, doi:10.3390/s20143841_

Round 1

Reviewer 1 Report

a) The first observation is, there is no code referenced in the article (https://github.com/cowbhave/SupplementaryMaterialsForCowLocalizationPaper), i.e., i suggest the authors to publish their code and data to the opensource community like GitHub to verify that article not is merely a README for a piece of software and is a research paper.

b) The article structure for the scientific novelty of a research article has no clear, it is recommended to divide it into the sections: 1.-Introduction, 2.-Theoretical Foundations, 3.-Materials and Methods, 4.-Tests and Results, 5.-Conclusion & Future Work, i.e., please add both physical and mathematical theoretical foundations for RFID geolocation among others on Materials and Methods section and correlate it with the experiments. 

c) Considering the alternatives existing today to improve positioning accuracy, there are inconsistencies between the RSSI-BLE functionality and the methodology proposed (e.g., it can be used location fingerprinting algorithm, particle filter algorithm, among other), so that the advantages and limitations with respect to other recent methodologies are not clearly appreciated. Thus, its recommended to add and contrast them (Viterbi algorithm) with other algorithms in the results and conclusions sections.

d) Finally, there is the question that correlates the pattern of the antennas and their influence on the location of objects, as well as defining the deformation of the near electromagnetic field and moving the antennas away from spatial orientation and the presence of objects and electromagnetic interferences in the use place.

Author Response

(x) Moderate English changes required

We made additional language check with a professional reviewer.

  1. a) The first observation is, there is no code referenced in the article (https://github.com/cowbhave/SupplementaryMaterialsForCowLocalizationPaper), i.e., i suggest the authors to publish their code and data to the opensource community like GitHub to verify that article not is merely a README for a piece of software and is a research paper.

We made the GitHub files public. Now they can be viewed.

  1. b) The article structure for the scientific novelty of a research article has no clear, it is recommended to divide it into the sections: 1.-Introduction, 2.-Theoretical Foundations, 3.-Materials and Methods, 4.-Tests and Results, 5.-Conclusion & Future Work, i.e., please add both physical and mathematical theoretical foundations for RFID geolocation among others on Materials and Methods section and correlate it with the experiments.

In this study we do not develop any physical and mathematical theoretical method, which must be described in a separate section. All the used methods are described in the Introduction in the context of review of existing methods, afterwards their specification for out localization method is detailed in the sections describing the methods, and their performance is compared to other methods in the Discussion. Nevertheless, we reorganized the structure of the paper. We replaced the general Methods section by section describing more specific topics: 2. Localization system and accuracy and 3.Tag RSS features.

In this study we concentrate on the localization methods based on BLE technology applied for the animal localization. Unfortunately, we have not found any study on RFID geolocation applied for the animal indoor localization.

  1. c) Considering the alternatives existing today to improve positioning accuracy, there are inconsistencies between the RSSI-BLE functionality and the methodology proposed (e.g., it can be used location fingerprinting algorithm, particle filter algorithm, among other), so that the advantages and limitations with respect to other recent methodologies are not clearly appreciated. Thus, its recommended to add and contrast them (Viterbi algorithm) with other algorithms in the results and conclusions sections.

We added explanations about the Viterbi algorithm in Lines 372-376: “The Viterbi algorithm was chosen among other filtering methods (fingerprint, particle filter, Kalman filter) because of the following reasons: it has low computation time; the location computation is based on the entire time interval, not only on the previous sampling; it uses additional information about cow behavior and the barn structure (which can be accessible for commercial barns); it does not require reference collecting for learning”.

  1. d) Finally, there is the question that correlates the pattern of the antennas and their influence on the location of objects, as well as defining the deformation of the near electromagnetic field and moving the antennas away from spatial orientation and the presence of objects and electromagnetic interferences in the use place.

The goal “estimate its accuracy” of this study was exactly to find features of the tags and the receiving stations for practical use in this application. We conducted additional experiment to find the pattern of the antennas and its deformation in the near electromagnetic field.

We added a description of the experiment in Lines 241-242: “To investigate the influence of the barn environment on the RSS-tag orientation dependence, an additional similar experiment with six tags and three receiving stations was conducted in the barn.” and in Lines 221-223 “The open space experiment was conducted to achieve the absolute features of the RSS tags and the receiving stations. In addition, we conducted barn experiments, which were intended to demonstrate a possible influence of the environment in specific conditions.”.

We added a description of the result in Line 300: “for the open space (a) and barn (b)” and Figure 8 (b).

We changed the conclusion in Lines 423-425: “This assumption corresponds with the RSS-tag orientation experiment presented in ‎Figure 8 (b), where the average deviation of the RSS for different orientations in the barn experiment is about 6 dB.”.

Reviewer 2 Report

The paper deals with the implementation of the localization system based on BLE signals in the barn to monitor the movement of dairy cows. 

The organisation of the paper should be improved, figures are placed quite far from the reference in the text, making readers jump around the paper. 

The description of the proposed and the implemented system should be described in more details. When reading thru the paper it is not quite clear what "mapping points" represent and what is their purpose in the system. This becomes clear later in the manuscript. 

Some information provided in the paper is useless since their impact on the system is questionable, e.g. what cows were fed with.

In equation (2) coefficients n and A0 are not described, what do these values represent? Only information provided in the manuscript says that their values were achieved from linear regression. 

What was the value of A0 used in the simulations? From the manuscript, it is not quite clear and it seems like different vales were used, which might not be the correct approach.  

Some figures are too complicated and hard to read, e.g. Figure 6 b), Figure 8, etc. 

"Figure 9. A typical cycle (5Hz) of tag current consumption including five data samplings and data package sanding."

"sanding" should probably be "sending"

It is not clear what is the relation of estimated position (blue) and probability in figure 11. I would expect the estimated position to be at the point with the highest probability, which is not the case. 

Comparison of results for different scenarios should be done under the same conditions. This is not the case since some results were achieved for "entire barn" and some for "waiting yard". However, these areas have different geometry and equipment, thus different signal propagation conditions. Therefore comparing these areas if the same propagation model parameters were used, might result in error introduced by propagation model optimized for one of the environments. 

What is the main advantage of the system compared to other systems described in the introduction of the paper?

Author Response

(x) Moderate English changes required

We made additional language check with a professional reviewer.

The paper deals with the implementation of the localization system based on BLE signals in the barn to monitor the movement of dairy cows.

The organisation of the paper should be improved, figures are placed quite far from the reference in the text, making readers jump around the paper.

We reorganized the structure of the paper: replaced the general Methods section by section describing more specific topics: 2. Localization system and accuracy and 3. Tag RSS features. Now all figures are located close to their first reference.

Multiple times the figures are used in the Discussion, though, we cannot duplicate the figures from the Results to the Discussion.

The description of the proposed and the implemented system should be described in more details. When reading thru the paper it is not quite clear what "mapping points" represent and what is their purpose in the system. This becomes clear later in the manuscript.

After the reorganization of the sections, the description of the mapping point is given before the further explanations about them.

Some information provided in the paper is useless since their impact on the system is questionable, e.g. what cows were fed with.

We deleted the redundant information (about the feed ratio, etc.).

In equation (2) coefficients n and A0 are not described, what do these values represent? Only information provided in the manuscript says that their values were achieved from linear regression.

We added an explanation taken from [31] in Lines 121-122: “n and A0 are the path loss exponent and the path loss constant”.

What was the value of A0 used in the simulations? From the manuscript, it is not quite clear and it seems like different vales were used, which might not be the correct approach. 

The values of the coefficients were given in Line 276: “A0= -48.77dB ± 4.070dB, n = 0.84 ± 0.119”.

We added explanation in Line 277: “This model (RSS = -10 ∙ 0.84 ∙ log(D) - 48.77) was further used in barn experiments.”.

Some figures are too complicated and hard to read, e.g. Figure 6 b), Figure 8, etc.

Instead of multiple thin lines we added boxplots to all sampling points and added captions to the Figures; “and RSS deviation (boxplots)” and “and the boxplots represent RSS deviation of the line with corresponding color”.

"Figure 9. A typical cycle (5Hz) of tag current consumption including five data samplings and data package sanding."

"sanding" should probably be "sending"

Line 293. Fixed.

It is not clear what is the relation of estimated position (blue) and probability in figure 11. I would expect the estimated position to be at the point with the highest probability, which is not the case.

We added in Lines 330-335: “Since the Viterbi algorithm for each sampling moment takes into account locations at the nearby moments (minimizing the total location error for the entire time interval), the calculated locations (circle) are not necessarily coincident with the points with the maximal location probability, which is not necessarily close to the actual tag location (asterisk) because of the high uncertainty of the RSS.”.

Comparison of results for different scenarios should be done under the same conditions. This is not the case since some results were achieved for "entire barn" and some for "waiting yard". However, these areas have different geometry and equipment, thus different signal propagation conditions. Therefore comparing these areas if the same propagation model parameters were used, might result in error introduced by propagation model optimized for one of the environments.

The RSS propagation experiment was conducted in relatively controlled environment, hence, the RSS propagation model achieved in this experiment was assumed to be the working model for the localization system. Both entire barn and waiting yard experiments are equally uncertain concerning application of the localization system, since the barn structure changing the propagation condition in both cases was too complicated for description and was not taken into account. The only controlled difference which was tested in these experiments was the influence of the average distance between the receiving stations and the localization accuracy. In this research we aimed only to demonstrate existing of this influence. Further investigation of this influence requires multiple experiments in different barn conditions, which was out of limitations of this research.

We improved the text to clarify this goal. We deleted confusing sentences in the first paragraph of the section 2.6. We added explanation in Lines 219-223: “The RSS propagation model (coefficients n and A0) based on data achieved from the open space experiment was set as the propagation model of the localization system and used in barn experiments. The open space experiment was conducted to achieve the absolute features of the RSS tags and the receiving stations. In addition, we conducted barn experiments, which were intended to demonstrate a possible influence of the environment in specific conditions.”.

What is the main advantage of the system compared to other systems described in the introduction of the paper?

We added to the Conclusions in Lines 251-256: “The advantages of the proposed localization system relative to existing commercial systems is in its low cost and long battery life. Unlike the majority of existing BLE based system research, the proposed system was designed for animal tracking in barn environment characterized by multiple objects reflecting and absorbing the BLE signal. Considering the similar systems [‎8], our system has better accuracy and RSS features of the tags in the barn environment were better studied.”

Reviewer 3 Report

The paper proposes a location system based on BLE that, although has low accuracy, is cheap compared with other solutions.

Nevertheless, there are important drawbacks that I think should have to be addressed in the paper:

1) It is not clear the accuracy needed: although the system is low cost and low accuracy, is that accuracy enough?

2) Authors characterize the system in a free zone, that gives information about the tags, but then, it is not clear how authors take into account the geometry of the barn.

3) It is not clear why authors need that low cost system for the cows. Are they only looking for the typical behavior of cows and detecting unusual behaviors? If that is the case, the results should have to be given in a different way to show that the system is able to detect uncommon patterns.

4) In the conclusions authors say that did some test taking into account the accelerometer and they were worse. Why not showing those results in the paper?

5) Finally, I think that a clarification is needed to show what the system brings to the body of knowledge of indoor positioning based on BLE.

On the other hand, there are some more specific elements that need to be addressed:

+ The link in the abstract does not work.

+ Space between numbers and unities are missed: 4.2m--> 4.2 m; 2m --> 2 m; etc.

+ m2 --> m^2

+ At the end of section 1 is missed the description of the structure.

+ Line 93: 9.8x42 m --> 9.8x42 m^2

+ Lines 114-115: here, I think that it is not well understood what authors mean, since they explain later that will take as position the closest point of the mapping. Maybe authors could explain the position system they will use at the very beginning of this section.

+ Lines 118-121: it is a repetition of the caption.

+ Line 132: less that several --> less than several.

+ Line 133: 300 s are 5 minutes. Even if cows changes rarely their location, 5 minutes seems a very long time to get the mean, mainly if the goal is to detect behaviors patterns (although I am not sure about that, as I said).

+ Line 158: authors speak about variability, but not about how obstacles are considered.

+ Lines 174-175: the role played by videos is not clear.

+ Line 198: what means "otherwise" here?

+ Figure 5: It is not clear to me the meaning of the percentages. Are they the percentage of every dB during 1 minute?

+ Also in Figure 5, authors say that a) is averaged and b) is filtered. Why this difference?

+ Figure 8 and, in general, the role played by orientation: it is not clear how it is taken into account.

+ Table 1: WS and q need to be defined.

+ Figure 10: What do colors mean? Suppose they are mean, median and Kalman, but they do not appear in the caption.

+ Figure 11a) It is not clear to me:  what this diagram represents. In two different time points, the blue dots are the same? In a single time, is always the same tag? The red point corresponds to the true position of the cow?

+ Figure 12: there are two elements, cows number 9, 26 and 27 with a very high errors. What are these high errors in these cows due to?

+ Figure 13: in the caption "c)" appears two times.

+ Line 341: one bracket is missed.

+ Line 394: can be the same.

+ Line 404: Authors say that: "The accuracy could be improved by usage of more advanced signal processing method, though, it is already close to the threshold of known possible accuracy for this technology". But I miss some justification of the algorithms they use, and why authors chose those. 

+ Lines 405 and 412: authors claim they get a low cost BLE system, but I miss a comparison with other BLE systems and their accuracy. 

Author Response

(x) English language and style are fine/minor spell check required

We made additional language check with a professional reviewer.

The paper proposes a location system based on BLE that, although has low accuracy, is cheap compared with other solutions.

Nevertheless, there are important drawbacks that I think should have to be addressed in the paper:

1) It is not clear the accuracy needed: although the system is low cost and low accuracy, is that accuracy enough?

The goal of the study was to estimate the accuracy of the system in a specific environment. This is needed for testing of its applicability for behavior recognition, which is the future plan of the study. Hence, we have not a required accuracy. We added in Lines 450-451: “The applicability of the system for recognizing unusual behavior will be tested in large scale experiments.”

2) Authors characterize the system in a free zone, that gives information about the tags, but then, it is not clear how authors take into account the geometry of the barn.

The structure of the barn influencing the RSS propagation was assumed as too difficult for description, hence, it was not taken into account in calculations with the RSS. To clarify it, we added in Lines 113-117: “Considering the large number of factors influencing the RSS in the barn environment (diversity of tag and receiving stations features, tag orientation, tag occlusions depending on cow movements, multiple metal structures in the barn), we assumed that RSS fingerprint mapping or using path loss models based on the barn structure description are impractical for this application. Hence, only the RSS propagation model in the free space was used.”.

Though, the barn structure was used in the location filtering by the Viterbi algorithm. To clarify it, we added in Line 163: “using the barn structure”.

3) It is not clear why authors need that low cost system for the cows. Are they only looking for the typical behavior of cows and detecting unusual behaviors? If that is the case, the results should have to be given in a different way to show that the system is able to detect uncommon patterns.

This is the next topic of the research. We added in Lines 450-451: “The applicability of the system for recognizing unusual behavior will be tested in large scale experiments.”.

4) In the conclusions authors say that did some test taking into account the accelerometer and they were worse. Why not showing those results in the paper?

Showing these results requires additional explanations about this method, which takes a lot of papers space and, actually, is useless, since it gave worse results. By reminding about the unsuccessful usage of accelerometers for defining the tag orientation we wanted to underline additional factor for the RSS uncertainty.

5) Finally, I think that a clarification is needed to show what the system brings to the body of knowledge of indoor positioning based on BLE.

We added to the Conclusions Lines 451-456: “The advantages of the proposed localization system relative to existing commercial systems is in its low cost and long battery life. Unlike the majority of existing BLE based system research, the proposed system was designed for animal tracking in barn environment characterized by multiple objects reflecting and absorbing the BLE signal. Considering the similar systems [‎8], our system has better accuracy and RSS features of the tags in the barn environment were better studied.”.

On the other hand, there are some more specific elements that need to be addressed:

+ The link in the abstract does not work.

We made the GitHub files public. Now they can be viewed.

+ Space between numbers and unities are missed: 4.2m--> 4.2 m; 2m --> 2 m; etc.

We fixed through the paper.

+ m2 --> m^2

Line . Fixed.

+ At the end of section 1 is missed the description of the structure.

We added in Lines 71-76: “The paper is organized as follows. Section ‎2 describes the structure of the localization system, experimental environment, methods used for the localization and conducted localization experiments. Section ‎3 describes the tag RSS features studied in the experiments. Section ‎4 presents the results of the experiments. Section ‎5 specifies the localization system accuracy, analyzes the experimental results and compares them with other studies. Section ‎6 summarize the findings and proposes plans for the future research.”

+ Line 93: 9.8x42 m --> 9.8x42 m^2

Line . Fixed.

+ Lines 114-115: here, I think that it is not well understood what authors mean, since they explain later that will take as position the closest point of the mapping. Maybe authors could explain the position system they will use at the very beginning of this section.

We changed the order of the sections, so that the mapping points are described in the beginning.

+ Lines 118-121: it is a repetition of the caption.

Deleted.

+ Line 132: less that several --> less than several.

Line 248. Fixed.

+ Line 133: 300 s are 5 minutes. Even if cows changes rarely their location, 5 minutes seems a very long time to get the mean, mainly if the goal is to detect behaviors patterns (although I am not sure about that, as I said).

We took that large value in purpose as an upper limit of the averaging window size range in the windows size optimization. However, majority of the time cows indeed don’t change their location: they lye for hours, eat for tens of minutes and stand without motion for minutes.

+ Line 158: authors speak about variability, but not about how obstacles are considered.

We added in Lines 113-117 a clarification for our assumption about the RSS propagation model that we use for the localization. However, the RSS-tag orientation experiment conducted in barn shows a possible influence of the obstacles on the RSS.

+ Lines 174-175: the role played by videos is not clear.

Line 152. We added “(required intensive manual work)”.

+ Line 198: what means "otherwise" here?

Line 176. We replaced it with “if the previous conditions are false”.

+ Figure 5: It is not clear to me the meaning of the percentages. Are they the percentage of every dB during 1 minute?

Line 265. Yes. We added “during the measurement time” to the Figure 5 caption.

+ Also in Figure 5, authors say that a) is averaged and b) is filtered. Why this difference?

Lines 267. We added “similar to the RSS propagation experiment” and “similar to the actual RSS measuring in barn” to the Figure 5 caption.

+ Figure 8 and, in general, the role played by orientation: it is not clear how it is taken into account.

The role of the orientation was critical, since it represents a significant factor for the RSS uncertainty. Hence, the influence of the tag orientation was analyzed in the study, and according to the results of the analysis we found that it should not be used in the proposed method. This decision was based on the speculative conclusion about the dependence of the RSS on the tag orientation in the barn environment that we made in Lines 395-396. Now (when we were able) we conducted an additional experiment on measuring the RSS-tag orientation in the barn condition, which approved our conclusion.

We added the motivation in Lines 233-234: “The diversity in the RSS values caused by the tag orientation represents a significant factor for the uncertainty in the tag localization, hence, it was studied carefully.”.

We added a description of the experiment in Lines 241-242: “To investigate the influence of the barn environment on the RSS-tag orientation dependence, an additional similar experiment with six tags and three receiving stations was conducted in the barn.”.

We added a description of the result in Line 300: “for the open space (a) and barn (b)” and Figure 8 (b).

We changed the conclusion in Line 423-425: “This assumption corresponds with the RSS-tag orientation experiment presented in ‎Figure 8 (b), where the average deviation of the RSS for different orientations in the barn experiment is about 6 dB.”.

+ Table 1: WS and q need to be defined.

Line 318. We added “with different window size (WS) and coefficient q for Kalman filter”.

+ Figure 10: What do colors mean? Suppose they are mean, median and Kalman, but they do not appear in the caption.

Lines 320. We added “(green – mean filter, black – median filter, purple – Kalman filter)” to the caption of Figure 10.

+ Figure 11a) It is not clear to me:  what this diagram represents. In two different time points, the blue dots are the same? In a single time, is always the same tag? The red point corresponds to the true position of the cow?

We rewrote the paragraph in Lines 327-334: “The performance of the Viterbi algorithm is demonstrated in four specific moments in ‎Figure 11 (b). The heat maps represent the probability of the tag localization calculated in each mapping point and based on the measured RSS, and the probability to move to other location points defined by the barn structure (equation ‎(4)). Since the Viterbi algorithm for each sampling moment takes into account locations at the nearby moments (minimizing the total location error for the entire time interval), the calculated locations (circle) are not necessarily coincident with the points with the maximal location probability, which is not necessarily close to the actual tag location (asterisk) because of the high uncertainty of the RSS”.

+ Figure 12: there are two elements, cows number 9, 26 and 27 with a very high errors. What are these high errors in these cows due to?

We found errors in the reference produced manually, some datasets were related to another cows. The cow number 9 which was wearing the tag number 5 has standing out upper whisker, though, its mean value is similar to the rest of the tags. We fixed the Figure 12 and total method accuracy (3.41 m ± 2.32 m).

+ Figure 13: in the caption "c)" appears two times.

Line 361. Fixed.

+ Line 341: one bracket is missed.

Line 343. Fixed.

+ Line 394: can be the same.

Line 420. Fixed.

+ Line 404: Authors say that: "The accuracy could be improved by usage of more advanced signal processing method, though, it is already close to the threshold of known possible accuracy for this technology". But I miss some justification of the algorithms they use, and why authors chose those.

We added a justification for choosing the Viterbi algorithm in Lines 372-376: “The Viterbi algorithm was chosen among other filtering methods (fingerprint, particle filter, Kalman filter) because of the following reasons: it has low computation time; the location computation is based on the entire time interval, not only on the previous sampling; it uses additional information about cow behavior and the barn structure (which can be accessible for commercial barns); it does not require reference collecting for learning.”.

We deleted this confusing sentence and added a paragraph containing a list of improvements in Lines 438-445: “Further improvement of the localization algorithm can be done in the following parts of the system. The RSS noise can be decreased by separation of the BLE signal channels as was done in [‎26] and [‎22]. The Viterbi algorithm can be improved by more detailed description of the cow behavior by the Markov chain, changing a method for calculation of the location probability in the mapping points and using additional information about the cow behavior measured by other sensors (such as cow body position estimated with the help of tag accelerometers) similar to [‎‎16]. To filter the RSS, a more advanced Kalman filter fitted to the RSS features or another filtering method can be found.”.

+ Lines 405 and 412: authors claim they get a low cost BLE system, but I miss a comparison with other BLE systems and their accuracy.

The comparison was in the Discussion section “such as 4.2 m at [‎8] in the cow barn environment and 2.4 m (for 90% of tested cases) at [‎22]”. We added additional references in Line 367: “3.8 m at [‎15] and 2.3 m at [‎20]”. The cost was not reported in the reviewed papers.

Reviewer 4 Report

The authors presente an interesting paper about cow location in barn based on low energy Bluetooth systems. however should be advised to implemente some improvements, namely:

1.- the references [20] and [21] are out of order.

2.- the presentation of the test conditions are not always easy to understand, so the authors should review the texto for this subject a litle.

3. - the part refering to data filtering, and its techniques, deserved better reasoning. For exemple, in relation to the Kalman Filter, there is only one test situation, the results of which are not all good. However, there is no great explanationTcomment about it. that is, the results are good/bad, why? What are de conditions/constraints lead to these results?

4. - The conclusions are very redutive. Basically the authors presente the erros of the system, without refering to percentages of the "match points" and, they give the idea that due to the level of erros they obteined the best is to transform the location system into a proximity system, that is, "detection of line to the milking robot". Also regarding the system improvements, the authors are very generic and vague. They just mention that the system can be improved by using methods developed by others. they do not mention what methods, and in particular how the improvements are expected to be.

Author Response

1.- the references [20] and [21] are out of order.

We fixed it.

2.- the presentation of the test conditions are not always easy to understand, so the authors should review the texto for this subject a litle.

We improved and organized the text. We divided the previous section 2 into: 2. Localization system and accuracy and 3. Tag RSS features. We added multiple explanations and complements. Please see throw the text.

  1. - the part refering to data filtering, and its techniques, deserved better reasoning. For exemple, in relation to the Kalman Filter, there is only one test situation, the results of which are not all good. However, there is no great explanationTcomment about it. that is, the results are good/bad, why? What are de conditions/constraints lead to these results?

We added to the discussion about the filters in Lines 430-431: “In this study, the simplest version of the Kalman filter unfitted to the RSS features was tested.”

  1. - The conclusions are very redutive. Basically the authors presente the erros of the system, without refering to percentages of the "match points"

The percentage of match with the reference points was given in the average cumulative accuracy presented in Figure 12 c and d.

and, they give the idea that due to the level of erros they obteined the best is to transform the location system into a proximity system, that is, "detection of line to the milking robot".

We assume that the further improvement of the localization system will provide accuracy comparable with the accuracy achieved in studies for the office environment, which already could be defined as the “localization” and not “proximity”.

Also regarding the system improvements, the authors are very generic and vague. They just mention that the system can be improved by using methods developed by others. they do not mention what methods, and in particular how the improvements are expected to be.

We concentrated the improvements in one paragraph in Lines 438-445: “Further improvement of the localization algorithm can be done in the following parts of the system. The RSS noise can be decreased by separation of the BLE signal channels as was done in [‎26] and [‎22]. The Viterbi algorithm can be improved by more detailed description of the cow behavior by the Markov chain, changing a method for calculation of the location probability in the mapping points and using additional information about the cow behavior measured by other sensors (such as cow body position estimated with the help of tag accelerometers) similar to [‎‎16]. To filter the RSS, a more advanced Kalman filter fitted to the RSS features or another filtering method can be found.”

Round 2

Reviewer 2 Report

Based on the text of the manuscript the parameters of the propagation model A0 and n were estimated based on measurements in the open space area and then implemented in the localization system deployed in the barn.

It is not clear why authors decided to use this approach, it might work for parameter A0 since it is dependent on transmit power and near space of the transmitter, however, parameter n is highly dependent on the environment in which the signal propagates. That might be the reason why in some cases the highest position probability is relatively far from the real position of the node.

Author Response

Based on the text of the manuscript the parameters of the propagation model A0 and n were estimated based on measurements in the open space area and then implemented in the localization system deployed in the barn.

It is not clear why authors decided to use this approach, it might work for parameter A0 since it is dependent on transmit power and near space of the transmitter, however, parameter n is highly dependent on the environment in which the signal propagates. That might be the reason why in some cases the highest position probability is relatively far from the real position of the node.

>>In Lines 115-116 we wrote that even in the open space environment, the uncertainty of the RSS measuring is so high, that using complicated propagation models is senseless, hence, we used the simplest one. In the open space experiment with a constant environment, the parameter n had a large variation indicating that there are uncontrolled factors influencing the simplest model, which will distort more complicated models even more. That was the reason why the real tag position was far from the highest position probability in some cases and was close in others, namely, the high uncertainty caused unpredictable errors.

Reviewer 3 Report

The paper has improved after the changes and now it is easier to understand. However, I still don't see what the paper brings to the state of the art from the technological point of view. 

Nevertheless, it is interesting to see a low cost application where location is get only for statistical purposes and precision plays a little rol.

In the attached document authors can find more specific comments.

Author Response

The paper has improved after the changes and now it is easier to understand. However, I still don't see what the paper brings to the state of the art from the technological point of view.

>>Indeed, the study does not have outstanding results. However, we achieved results better than in a similar study of Trogh et al. (2018), which, actually, was the single study about the BLE localization in the indoor agricultural environment that we succeeded to find. In addition, we performed a detailed study of the BLE tag features and a specific type of environment, which had not been studied enough, and which represents a typical environment in agricultural applications. All this was mentioned in Lines 453-456.

Nevertheless, it is interesting to see a low cost application where location is get only for statistical purposes and precision plays a little rol.

>>To the applications mentioned in Lines 449-451 we added “detecting abnormally long walks during heating”.

In the attached document authors can find more specific comments.

Line 85. But then, accelerations due to head movements are also recorded.

>>Yes, but the acceleration is not a subject of this study. We mention it only to emphasize that the proposed localization system is low-cost, since it uses all the equipment which is anyway used by the system measuring the acceleration.

Line 117. Don't see why fingerprinting is impractical. Metalic structures do not move.

>>The main reasons for this are written in Lines 391-394: “high RSS variability, uncertain factors such as blocking the signal by other cows, or cows bending necks”. The high RSS variability includes not only the variability between the tags requiring creating the fingerprint map for all tags separately, but also orientational variability multiplying the number of the sampling points in the fingerprint map by the number of sampled orientations for each tag.

On the other hand, it is not clear to me what authors consider free space.

>>Thank you for the comment. We changed it to “created in a tag feature experiment (described in ‎3.1)”.

Line 120. The symbol for log must be in roman.

>>Fixed.

Line 120. But precisely due to the metal elements and the structure of the arn, this model is not reliable when there are obstacles.

>>Yes, the model does not describe exactly the RSS propagation in the barn. But, as we wrote in Lines 113-116, the environment is so complicated, that it is senseless to use more advanced propagation models: with much harder efforts we doubt we can gain any improvement.

Line 143. They are not normalization factors, but "smoothing" factors.

>>We added in Line 139 “we canceled this difference by the shifting factors” and changed it to “shifting factor” through the entire paper.

Line 250. The power should be linked to the number.

>>Fixed.

Line 264. What do these percentages mean? Is the percentage of time that every signal value appears?

>>We added in Line 265 “frequency distribution in histogram” and added “Frequency” to the caption of the histograms in Figure 5.

Line 349. Even if this is the mean value, we have a value with an error higher than 5 m. Is there any reason for that error?

>>We found additional errors in the manual reference and fixed Figure 12 and the system accuracy accordingly: 3.27 m ± 2.11. Thank you.
